# The Prognostic Role of CDK9 in Bladder Cancer

**DOI:** 10.3390/cancers14061492

**Published:** 2022-03-15

**Authors:** Jędrzej Borowczak, Krzysztof Szczerbowski, Mateusz Maniewski, Marek Zdrenka, Piotr Słupski, Paulina Antosik, Sylwia Kołodziejska, Marta Sekielska-Domanowska, Mariusz Dubiel, Magdalena Bodnar, Łukasz Szylberg

**Affiliations:** 1Department of Clinical Pathomorphology, Collegium Medicum in Bydgoszcz, Nicolaus Copernicus University in Torun, 85-094 Bydgoszcz, Poland; szczerbowskikm@gmail.com (K.S.); mm.maniewski@gmail.com (M.M.); paulina.antosik@cm.umk.pl (P.A.); magdabodnar@o2.pl (M.B.); l.szylberg@cm.umk.pl (Ł.S.); 2Department of Tumor Pathology and Pathomorphology, Oncology Centre—Prof. Franciszek Łukaszczyk Memorial Hospital, 85-796 Bydgoszcz, Poland; zdrenka.marek@gmail.com; 3Department of Urology, University Hospital No. 2 im. Dr. Jan Biziel in Bydgoszcz, 85-168 Bydgoszcz, Poland; piotr.slupski@icloud.com; 4Chair of Pathology, University Hospital No. 2 im. Dr. Jan Biziel in Bydgoszcz, 85-168 Bydgoszcz, Poland; sylwia15913@wp.pl; 5Department of Obstetrics, Gynecology and Oncology, Ludwik Rydygier Collegium Medicum in Bydgoszcz, Nicolaus Copernicus University in Toruń, 85-168 Bydgoszcz, Poland; marta.sekielska@gmail.com (M.S.-D.); dubiel@cm.umk.pl (M.D.)

**Keywords:** CDK9, bladder cancer, expression, prognosis, survival

## Abstract

**Simple Summary:**

In this article, we investigated the prognostic role of cyclin-dependent kinase 9 expression in urothelial carcinoma. High CDK9 expression has recently been associated with shorter patient survival time, but its role in urothelial carcinoma has not yet been explored. The expression of CDK9 was higher in cancer than in normal urothelial tissue and correlated with tumor grade, stage, and invasiveness. To our surprise, patients with high CDK9 expression lived longer than patients with low CDK9 expression. In The Cancer Genome Atlas database cohort, high CDK9 RNA concentration correlates with longer survival of patients and CDK9 status remained a statistically significant prognostic factor in multivariate analysis. It seems that CDK9 not only regulates the expression of anti-apoptotic genes, leading to longer survival of cancer cells, it also facilitates DNA repair, preventing the build-up of genomic instability, crucial in the initiation and progression of bladder cancer. The results suggest that CDK9 overexpression is not always associated with a worse prognosis, while cell maturity and disease stage may influence the efficacy of potential targeted therapy.

**Abstract:**

Introduction: Most patients with urothelial carcinoma are diagnosed with non-invasive tumors, but the prognosis worsens with the progression of the disease. Overexpression of cyclin-dependent kinase 9 has been recently linked to increased cancer proliferation, faster progression, and worse prognosis. However, some cancers seem to contradict this rule. In this work, we explored the prognostic role of CDK9 expression in urothelial carcinoma. Materials and Methods: We performed immunohistochemical analysis on 72 bladder cancer samples. To assess a larger group of patients, the Cancer Genome Atlas (TCGA) database containing 406 cases and transcriptomics information through the Human Pathology Atlas were analyzed. Results: CDK9 is overexpressed in urothelial cancer tissues when compared to normal urothelial tissues (*p* < 0.05). High CDK9 expression was observed in low-stage, low-grade, and non-muscle-invasive tumors (*p* < 0.05). The patients with high CDK9 expression had a significantly higher 5-year overall survival rate than those with low CDK9 expression (77.54% vs. 53.6% in the TMA group and 57.75% vs. 35.44% in the TCGA group, respectively) (*p* < 0.05). The results were consistent in both cohorts. Multivariate Cox regression analysis indicated that low CDK9 status was an independent predictor for poor prognosis in the TCGA cohort (HR 1.60, CL95% 1.1–2.33, *p* = 0.014). Conclusions: High CDK9 expression predicts a favorable prognosis in urothelial carcinoma and is associated with clinicopathological features characteristic for early-stage disease. The decrease in CDK9 expression can be associated with the build-up of genetic instability and may indicate a key role for CDK9 in the early stages of urothelial carcinoma.

## 1. Introduction

Bladder cancer is the ninth most frequent malignancy worldwide, with approximately 430,000 cases a year. It ranks 13th in terms of mortality with almost 200,000 deaths per year [1,2,3]. The most common histological subtype is urothelial cancer, which accounts for approximately 90% of the cases. Over half of the patients are diagnosed in the early stage of the disease with non-invasive tumors and are successfully treated radically. This results in a high 5-year survival rate of up to 77.1%. However, when the disease is more advanced, these numbers drop dramatically, reaching a 4.7% survival rate in metastatic cancer [2]. Genetic heterogeneity, reactive increase in DNA repair, and mechanisms modifying the intracellular drug concentration may limit the response to therapy [4]. Therefore, there is a need for novel treatment options as well as novel prognostic markers.

### Cyclin-Dependent Kinase 9 (CDK 9)

Cyclin-dependent kinase 9 (CDK9) is a transcription regulating protein [5]. Together with cyclin T, CDK9 forms positive transcription elongation factor-B (P-TEFb), which activates RNA polymerase II (RNA POL II), and through this mechanism, stimulates transcription [5,6]. The following translation results in the formation of anti-apoptotic proteins, such as MYC or Mcl-1 [7]. This disrupts cellular homeostasis, shifting the apoptotic balance towards the survival of cells [8]. At the same time, the recruitment of P-TEFb is required for the differentiation of muscles [9], neurons [10], or adipocytes [11]. Furthermore, CDK9 promotes tumor growth via the p53 related pathway [12,13]. Its overexpression is associated with poor prognosis in various neoplasms, such as pancreatic cancer and osteosarcoma [14,15]. CDK9 inhibitors are being tested for the treatment of multiple malignancies, including multiple myeloma, acute myeloid leukemia, prostate cancer, and hepatocellular carcinoma, making CDK9 a valid potential therapeutic target and a novel prognostic marker [16,17,18,19]. In this work, we aimed to investigate whether there is a connection between the CDK 9 expression and individual clinical features of bladder cancer, such as stage, grade, presence of metastasis, and survival time. We assessed the prognostic value of CDK9 expression in urothelial cancer and validated the findings in The Cancer Genome Atlas Program database.

## 2. Materials and Methods

### 2.1. Patients and Tissue Samples

All tissue specimens were collected from patients diagnosed with urothelial carcinoma and treated in the Department of Urology between November 2009 and July 2018. Our study includes 72 cases of bladder cancer (study group) and 32 cases of normal urothelial mucosa (control group), collected immediately during either transurethral resection of bladder tumor (TURBT) or radical cystectomy (RC). Clinical data, including age, sex, overall survival, tumor differentiation (grade), stage T, lymph nodes invasion, metastasis, tumor size, cancer invasiveness, progression, and recurrence were obtained (Table 1). The study was conducted following the Declaration of Helsinki, and the protocol was approved by the Bioethics Committee (KB881/2019).

### 2.2. Sample Staining

The expression of CDK9 was determined using IHC assays according to the protocol described in Buchholz et al.’s study [20]. In the beginning, standardization and optimization of the IHC method were performed on a recommended tissue, based on the antibody datasheet and reference sources (The Human Protein Atlas: https://www.proteinatlas.org (accessed on 11 November 2021); Uhlen et al., 2010 [21]). In brief, 3 μm thick sections of the tissue arrays were baked for 1 h at 60 °C before xylene deparaffinization and subsequent rehydration through graded ethanol (99.8%; 96%; 90% and 80%). Tissue sections were incubated with a primary rabbit monoclonal anti-CDK9 antibody (1:200; 40 min; cat. no: ab76320, Abcam, Cambridge, MA, USA). Primary antibodies were visualized using the UltraView Universal DAB Detection Kit (Roche Diagnostics/Ventana Medical Systems, Tucson, AZ, USA) followed by color development using 3,3-diaminobenzidine. The slides were counterstained with hematoxylin II for 12 min and blue reagent for 4 min. Finally, tissue sections were dehydrated in increasing ethanol concentrations (80, 90, 96, and 99.8%), cleared in xylenes (I–IV), mounted using a mounting medium, and examined.

### 2.3. Image Acquisition and IHC Analysis

Initially, the clinical data were blinded and the images were captured using an optical microscope at ×10 magnification with a color video camera attached to a computer system. For each sample, two experienced pathologists selected the most representative regions and acquired images. The analysis was performed using the ImageJ 1.53j version (NIH, Bethesda, MD, USA) (Java 1.8.0_172) and the IHC profiler plugin. Nuclear CDK9 expression was obtained by calculating the H-score. To determine CDK9 expression in cancer cells, the standard protocol designed by Verghese et al. was followed [22]. The highly positive zone was found to be ranging from 1 to 60; 61 to 120 for the positive zone; 121 to 170 for the low positive zone; and 181 to 220 for the negative zone, respectively. The intensity values ranging from 221–255 predominantly represent fatty tissues, stroma, or background artifacts that do not contribute to pathological scoring and were therefore excluded from the score determination zones. H-score was assigned using the formula (1 × (%cells low positive) + 2 × (%cells positive) + 3 × (%cells high positive)), obtaining a value from 0 to 300.

### 2.4. Statistical Analysis

All statistical analyses were performed using Statistica version 13.3 (Statsoft) and Microsoft Excel 2019. The value of *p* < 0.05 was considered statistically significant. Continuous variables were tested for normality by the Kolmogorov–Smirnov test. The relations between compared groups, due to the categorical character of variables, were analyzed using the Mann–Whitney U Test. More than two independent groups were compared using the ANOVA Kruskal–Wallis test. Correlation between clinicopathological characteristics and CDK9 expression was evaluated using Spearman’s rank correlation coefficient. Univariate and multivariate analyses of potential predictors for overall survival were performed using Cox proportional hazard regression. Results were expressed as hazard ratio (HR) and 95% confidence interval (CI). The two-sided *p*-value of <0.05 was considered to indicate statistical significance. The relation between CDK9 expression with overall survival was evaluated with a log-rank test and presented using Kaplan–Meier analysis.

## 3. Results

### 3.1. Patients Characteristics

We explored the relevance of CDK9 expression in human urothelial carcinoma by comparing normal urothelial mucosa and urothelial carcinoma of bladder cancer patients. Table 1 summarizes the characteristics of the TMA cohort. The research group consisted of 11 females and 61 males. The mean age of patients was 71.5 years (range 45–88 years) and the median follow-up time was 5 years. Among 72 patients, 34 (47.22%) were diagnosed with low-grade tumors and 38 (52.78%) were diagnosed with high-grade tumors. 39 (54.17%) tumors were classified as T1, 20 (27.78%) as T2, 9 (12.5%) as T3, and 4 (5.56%) as T4. The samples were categorized as low stage (T1) or high stage (T2-4). Nine (12.5%) patients were diagnosed with lymph node metastases and seven (9.72%) patients had distant metastases at the time of diagnosis. The mean 5-year overall survival time was 45.3 months, ranging from 5.0 to 60.0 months.

### 3.2. CDK9 Is Overexpressed in Bladder Cancer

To explore the characteristics of CDK9 staining patterns in urothelial cancer and control samples, we performed immunohistochemical staining using a monoclonal CDK9 antibody (1:200; 40 min; cat. no: ab76320, Abcam, Cambridge, MA, USA). CDK9 expression was present in all examined samples in both study and control groups. Strong immunoreactivity was observed in bladder cancer samples and was significantly higher than in the control group (median H-SCORE = 204 vs. 170.5 respectively, *p* = 0.0022) (Figure 1). CDK9 is overexpressed in urothelial carcinoma.

### 3.3. CDK9 Expression Correlates with Disease Course in Bladder Cancer TMA Cohort

According to the Mann–Whitney U test, CDK9 expression was significantly higher in the lower stage (pT1 vs. pT2–4; *p* = 0.0172), lower grade (low vs. high; *p* = 0.04), and non-invasive tumors (NMIBC vs. MIBC; *p* = 0.0075) (Figure 2). The detailed description of CDK9 expression in selected groups is assembled in Table 2. CDK9 expression in T1 tumors was significantly higher than in the T2–T4 group and in the control. However, we found no significant difference between CDK9 expression in the T2–T4 group and the control. Spearman’s correlation coefficient showed a weak to moderate negative correlation between CDK9 expression and tumor stage, grade, size, and invasiveness (*p* < 0.05). CDK9 expression did not correlate with metastasis, lymph node invasion, recurrence, or progression of the disease (*p* > 0.05) (Table 2).

To determine the prognostic value of CDK9 expression in patients with urothelial carcinoma, we dichotomized the samples into low and high CDK9 expression groups, with the cutoff point being 219 H-score. Patients with high CDK9 expression had a significantly higher 5-year overall survival (OS) rate than patients with low CDK9 expression (77.54% vs. 53.6%, respectively; *p* = 0.04) (Figure 3). The Kaplan–Meier analysis of OS by quartiles showed significant differences in OS between patients in the lower and upper quartiles of CDK9 expression (*p* = 0.039) (Figure 3).

Univariate Cox regression analysis showed that the type of procedure, stage, grade, invasiveness, tumor size, lymph node invasion, presence of distant metastases, and progression were significant prognostic factors. In multivariate analysis, only the occurrence of progression remained statistically significant (<0.05) (Table 3). CDK9 status was not statistically significant for the prognosis of overall survival (HR 2.7, CI95% 0.93–7.82, *p* = 0.06), but due to borderline statistical significance and small group size, we decided to explore the prognostic value of CDK9 in the TCGA urothelial cancer cohort.

### 3.4. TCGA Urothelial Bladder Cancer Cohort

We found that CDK9 expression correlates with higher OS rate, lower stage, and grade (Figure 2, Table 2). However, due to a relatively small number of cases, with only 13 tumors being T3 or T4, as well as a lack of statistical significance in Cox regression analysis and contradictory reports from other researchers, we deemed it necessary to validate our findings. To assess a larger group of patients, we accessed The Cancer Genome Atlas (TCGA) database and obtained transcriptomics information through the Human Pathology Atlas. The TCGA cohort consisted of 406 cases with urothelial bladder cancer, out of which 273 samples were high stage (T3 or T4) (Table 4) [23]. The Ensembl gene id, available from TCGA, was used to map the TCGA RNA-seq data and the FPKMs (number of fragments per kilobase of exon per Million reads). Based on the FPKM value of CDK9, the samples were dichotomized into the low expression and high expression groups. Univariate Cox regression analysis showed that age, stage, and CDK9 status were statistically significant prognostic factors. All predictors, including CDK9 status (HR 1.60, CL95% 1.1–2.33, *p* = 0.014) remained significant in multivariate analysis (Table 5). The 5-year survival rate in patients with high CDK9 expression reached 57.75% and was significantly higher than the 35.44% 5-year survival rate in the low CDK9 expression group (*p* < 0.005) (Figure 4). We found no differences between patients’ OS by quartiles in the Kaplan–Meier analysis (Figure 4). The results obtained from the TCGA cohort are consistent with the findings in the TMA cohort.

CDK9 expression was prognostic in the TCGA cohort, and its high expression predicts longer overall survival in urothelial bladder cancer.

## 4. Discussion

### 4.1. The Prognostic Role of CDK9 in Cancers

The presented results show that although CDK9 is overexpressed in all stages and grades of bladder cancer when compared to normal bladder tissue, its expression decreases in line with higher grade and stage. With the Kaplan–Meier estimator, the results indicate that CDK9 may predict a good prognosis in patients with bladder cancer. However, available literature mentions an ambiguous prognostic role of CDK9, which differs in various types of cancers.

Kretz et al. [15] showed that CDK9 is overexpressed in pancreatic ductal adenocarcinoma and higher CDK9 expression correlates with shorter survival times in PDAC patients. In Ma et al.’s study, CDK9 expression is inversely correlated to the percent of tumor necrosis post-neoadjuvant chemotherapy, an important predictive factor for disease outcomes in osteosarcoma patients and correlates with worse prognosis [14]. Wang et al. reported that in ovarian cancer high-CDK9 expression correlated with significantly shorter overall survival time and disease-free survival. CDK9 expression was also significantly higher in the patient-paired metastatic and recurrent tissue when compared to primary ovarian cancer tissue [24]. Similarly, Parvathareddy et al. reported that high CDK9 expression is an indicator of poor prognosis, tumor recurrence, and high Ki-67 index in epithelial ovarian cancer (441 samples) [25]. On the other hand, Schlafstein et al.’s study revealed that high CDK9 expression was associated with longer overall survival starting in patients at 3 years after the initial surgery, who did not achieve complete response after neoadjuvant chemotherapy [26].

### 4.2. CDK9 in Cell Differentiation and Carcinogenesis

Berthet and Kaldis [27] suggested that well-differentiated cells are more sensitive to cell cycle dysregulation. In mouse models, cell cycle regulating mechanisms were different in embryonic stem cells and differentiated cells [28]. When tumor cells proliferate, they start to behave similarly to stem cells. However, inhibition of CDK can affect proliferation, and in this scenario, tumor cells behave more like differentiated cells. It seems plausible that in specific tumors, CDK overexpression may be pivotal especially in the early stages of disease when relatively well-differentiated cells are more dependent on Cdk/cyclin complexes and the genomic instability is still limited. Schlafstein et al. drew similar conclusions, arguing that if low CDK9 expression leads to increased DNA damage and genetic instability, then the disease in patients with low CDK9 expression may be more aggressive [26,29]. This thesis seems to be reinforced by De Falco et al.’s study, where higher expression of CDK9 was observed in PNET and neuroblastoma tumors with more differentiated cells [10].

In our study CDK9 expression was the highest in the low-stage tumors, suggesting that CDK9 overexpression may play an important role in cancer development, but its role decreases when the genomic instability increases (Figure 2). The main implications of CDK9 overexpression are summarized in Figure 5 [12,13,30]. Similarly, CDK9 expression was higher in the low-grade group than in the high-grade group, which can be attributed to the increasing independence from cell cycle regulators in higher-grade cancers and suggest that the role of CDK9 may be marginalized as the disease progresses [31] (Figure 2).

### 4.3. CDK9 and Genome Stability

Yu et al. suggested that CDK9 plays a key role in maintaining genome integrity in response to replication stress [32]. CDK9 silencing in U2OS cells resulted in delayed progression through S-phase. Similar results were observed in human telomerase-immortalized epithelial cells, suggesting that the effects are independent of cell type. The recovery defect was similar to that after treatment of aphidicolin, a DNA polymerase inhibitor. In the absence of exogenous damage, CDK9-silencing caused no changes in proliferation and apoptosis. The induction of DNA damage after CDK9 knockdown led to replication fork instability and breakdown, even in the absence of added genotoxic agents [32,33,34]. Those findings suggest that CDK9 is needed to complete DNA synthesis and contributes to maintaining genome integrity in a response to replication stress. Interestingly, only the deficit in cyclin K, but not cyclin T1 or cyclin T2, impaired the cell cycle recovery, suggesting that cyclin K is the regulatory subunit of CDK9, which mediates its activities in the RSR [32].

Low-grade bladder cancers are usually non-muscle-invasive, but due to the increasing genomic instability, they may progress to invasive tumors [35,36]. In Vaish et al. study cancer microsatellite instability was observed frequently in high-stage (40.6%) and high-grade (59.4%) tumors [36]. CDK9 accumulates in response to replication stress and lifts the burden of transcriptional stress by limiting the amount of single-stranded DNA in cells. CDK9 knockdown increases the spontaneous DNA damage signaling in replicating cells and impairs their ability to recover from a transient cell cycle arrest [32]. Relatively high CDK9 expression in the non-muscle-invasive and low-grade cancer groups (Figure 2) seems to be in line with those reports.

The p53 human suppressor gene plays a pivotal role in maintaining genomic stability by regulating the cell cycle, cell differentiation, DNA repair, and apoptosis [37]. The loss of p53 function is associated with lower overall survival of bladder cancer patients and is the most prevalent in high-grade, high-stage, and muscle-invasive cancers [38,39,40,41]. CDK9 and p53 form a feedback loop, in which CDK9 phosphorylates p53 and renders its ability to cause cell cycle arrest and apoptosis, while p53 increases CDK9 gene expression [42]. In response to DNA damage, p53 also activates the transcription of cyclin K, critical for genomic maintenance and replication shock response. As cyclin K and cyclin T differ structurally, the CDK9-cyclin K complex can act independently of cyclin T [29,43]. The reduction in CDK9 activity may be a direct consequence of p53 mutation, its subsequent loss of function, and dysregulation of the p53-CDK9 feedback loop. In that sense, CDK9 expression may be an indicator of p53 functionality. We hypothesize that in low-grade urothelial cancer, CDK9 overexpression may diminish p53 activity, facilitating progression. However, in high-grade tumors, p53 is usually mutated or inactive, therefore the relatively reduced expression of CDK9 would limit the stabilizing activity of cyclin K and further impair DNA repair mechanisms [44].

The pathways between CDK9, p53, and other tumor-suppressive proteins are well established, but it is still unclear whether the decrease in CDK9 activity arises from their interactions. The assumption that the relationship between CDK9 and other proteins differs in nature when compared to other malignancies seems doubtful. The relatively decreased CDK9 expression may be a direct manifestation of genomic instability. The most frequent genetic alteration in transitional cell carcinoma is the loss of chromosome 9, occurring in >50% of bladder tumors for all grades and stages [45]. Deletions of chromosome 9 more frequently affect 9q than 9p and are more prevalent in higher-grade tumors [46,47]. Tumors with deletions of the regions 9ptr-p22, 9q22.3, 9q33, and 9q34 recur more rapidly than those without deletion [47]. Loss of heterogeneity of 9q is considered a very early genomic alteration in bladder cancer pathogenesis and the most common event amongst a series of copy number changes, suggesting that loss of 9q leads to a rapid increase in genomic instability [45]. Since the CDK9 gene is located on chromosome 9q34, it is possible that the decrease in CDK9 expression is a result of CDK9 knockdown in genetically unstable cells and reflects the destabilization of the genome [48,49]. According to this hypothesis, in cancers with a relatively lower frequency of somatic mutations, the expression of CDK9 is not hindered by genomic instability and high CDK9 expression may predict a poor prognosis. However, in cancers where somatic mutations are more frequent, the genomic instability, including early deletions of 9q, may decrease the expression of CDK9. This statement seems to be true for bladder cancer and lung cancer, which are characterized by a high frequency of somatic mutations and in which low CDK9 expression correlates with a shorter overall survival time [50,51]. In those tumors, low CDK9 expression may be an indicator of more aggressive disease.

Low CDK9 expression in urothelial cancer tissues correlates with more advanced, higher-grade, and muscle-invasive disease, therefore subjecting low-expression patients to more aggressive therapy may provide clinical benefits. DNA repair gene mutations are prevalent in this group; therefore, combined therapy of CDK9 inhibitors with other agents that impair DNA repair, such as PARP inhibitors, may be beneficial [52]. However, co-inhibition of CDK9 and PARP has yet to be proven in urothelial cancer cell lines [45]. Furthermore, CDK9 silencing resulted in no modification of DNA repair genes in SAS and FaDu cells, suggesting another mechanism of action [29,46,53]. On the other hand, in the early stage of urothelial cancer, where CDK9 expression is the highest, CDK9 inhibition can inhibit transcription of anti-apoptotic proteins, impair tumor growth, reactivate wild-type p53 and increase its concentration, thereby preventing disease progression [12,47,54]. As CDK9 has yet to be investigated as a therapeutic target in urothelial carcinoma, preclinical studies should be performed before attempting clinical trials.

## 5. Conclusions

Higher CDK9 expression correlates with a lower grade, lower stage, and non-muscle-invasive bladder cancer. Urothelial bladder cancer patients with higher CDK9 expression had a higher 5-year overall survival rate when compared to the low CDK9 expression group. Contrary to results from other malignancies, CDK9′s role in bladder cancer seems different. Its high expression seems to be more significant in low-stage tumors, where p-53 mutations are rare and the genome is stable. Along with the increase in genomic instability, CDK9 decreases due to a decrease in p53 functionality, deletions of chromosome 9q, or dedifferentiation of cancer cells. Although our findings suggest that the CDK9 influence on disease progression is not clearly negative, there are no proven mechanisms that would confirm CDK9′s duality in carcinogenesis. The ambiguous role of CDK9 needs further evaluation.

## Figures and Tables

**Figure 1 cancers-14-01492-f001:**
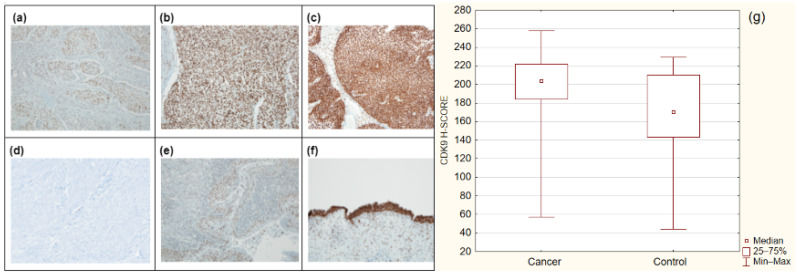
Representative cross-sectional staining patterns at x10 magnitude; (**a**) invasive bladder cancer with low CDK9 expression; (**b**) non-invasive bladder carcinoma with medium CDK9 expression; (**c**) non-invasive bladder cancer with high CDK9 expression; (**d**) CDK9 negative control; (**e**) normal mucosa with low CDK9 expression; (**f**) normal mucosa with high CDK9 expression and positive reaction in the cells of the stromal inflammatory infiltration; (**g**) CDK9 expression in cancer and control groups (*p* = 0.0022).

**Figure 2 cancers-14-01492-f002:**
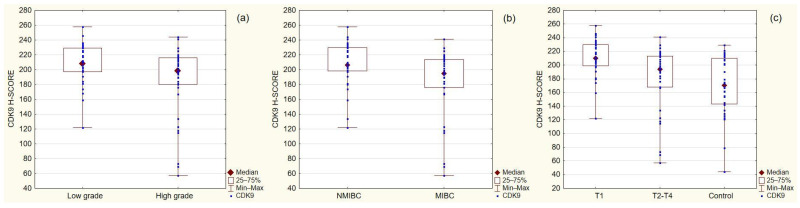
CDK9 expression depending on: (**a**) tumor grade (*p* = 0.04); (**b**) tumor invasiveness (*p* = 0.0075); (**c**) tumor stage (*p* = 0.0001). NMIBC—non-muscle-invasive bladder cancer; MIBC—muscle-invasive bladder cancer.

**Figure 3 cancers-14-01492-f003:**
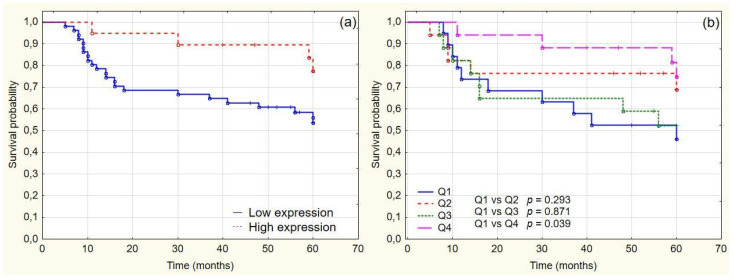
OS analysis of TMA cohort (**a**) overall survival in low CDK9 and high CDK9 groups (5-year OS 77,54% vs. 53.6%, respectively; *p* = 0.04); (**b**) overall survival by CDK9 expression quartiles.

**Figure 4 cancers-14-01492-f004:**
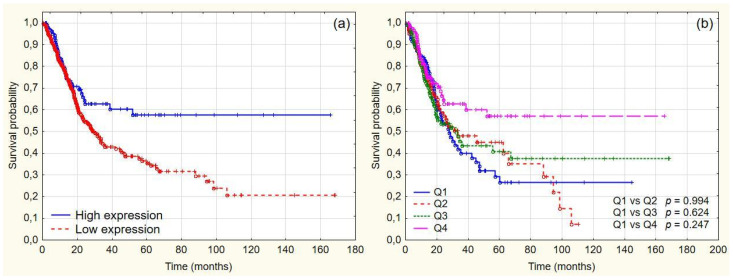
Overall survival analysis of TCGA cohort (**a**) overall survival in low CDK9 and high CDK9 groups (5-year OS 57.75% vs. 35.44%), respectively (*p* = 0.009); (**b**) overall survival by CDK9 expression quartiles.

**Figure 5 cancers-14-01492-f005:**
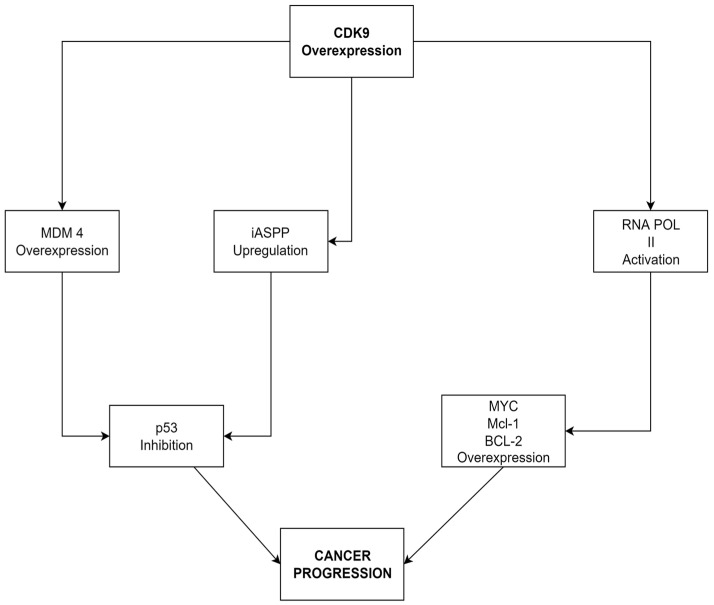
The role of CDK9 overexpression in cancer progression. CDK9 activates the mouse double minute 4 (MDM4) and the inhibitor of apoptosis-stimulating protein of p53 (iASPP) proteins, which inhibit the tumor-suppressing functions of p53 and disturb genomic integrity. Overexpression of CDK9 increases the activity of RNA polymerase II and causes an increase in anti-apoptotic proteins, such as c-Myc, Mcl-1, and Bcl-2, preventing the programmed death of cancer cells. Both pathways lead to the accumulation of genetic changes and the progression of the disease [12,13,30].

**Table 1 cancers-14-01492-t001:** Clinicopathological characteristics of the study group.

Variables		*n* (%)
Age	Mean	71.5 years (range 45–88 years)
Sex	Female	11 (15.28%)
	Male	61 (84.72%)
Grade	low	34/72 (47.22%)
	high	38/72 (52.78%)
Stage	T1	39/72 (54.17%)
	T2	20/72 (27.78%)
	T3	9/72 (12.5%)
	T4	4/72 (5.56%)
Tumor size	≥3 cm	39/72 (54.17%)
	<3 cm	33/72 (45.83%)
Lymph node metastases	N0	61/72 (84.72%)
	N1–3	9/72 (12.5%)
	Unknown	2/72 (2.78%)
Distant metastasis	No	62/72 (86.11%)
	Yes	7/72 (9.72%)
	Unknown	3/72 (4.17%)
Invasiveness	NMIBC	36/72 (50%)
	MIBC	35/72 (48.61%)
	Unknown	1/72 (1.39%)
Progression	Yes	17 (23.61%)
	No	34 (47.22%)
	Unknown	21 (29.17%)
Recurrence	Yes	26 (36.11%)
	No	8 (11.11%)
	Unknown	38 (52.78%)
Mean recurrence time		21.07 months
Type of procedure	TURBT	35 (48.61%)
	PC	31 (43.06%)
	Unknown	6 (8.33%
Disease course	Alive	29/72 (40.28%)
	Dead	43/72 (59.72%)
Median follow-up time		60 months (range 5–60 months)

**Table 2 cancers-14-01492-t002:** Correlation between CDK9 expression and clinical predictors for bladder cancer.

Clinical Data	Total *N*	Median CDK9 Expression (Min–Max)	Q1	Q3	Statistical Differences between Groups (*p* < 0.05)	CDK 9 Expression Correlation (Spearman’s Correlation Coefficient)
Cancer group	72	204 (57–258)	184	222	-	
Low grade	34	208 (122–258)	197	229	*p* = 0.04	−0.283 (*p* < 0.05)
High grade	38	198.5 (57–244)	180	216		
T1	39	210 (122–258)	199	230	*p* = 0.0001	−0.35 (*p* < 0.05)
T2–T4	33	194 (57–241)	168	213		
NMIBC	36	206 (122–258)	198	229	*p* = 0.0075	−0.34 (*p* < 0.05)
MIBC	35	195 (57–241)	176	214		
N0	61	206 (57–258)	190	225	*p* = 0.31	0.05 (*p* > 0.05)
N1-3	9	184 (69–246)	167	218		
M0	62	204.5 (57–258)	181	224	*p* = 0.91	0.026 (*p* > 0.05)
M1	7	205 (69–244)	184	225		
Progression	17	195 (69–244)	197	226	*p* = 0.19	0.16 (*p* > 0.05)
Lack of progression	35	210 (115–258)	184	212		

NMIBC—non-muscle-invasive bladder cancer; MIBC—muscle-invasive bladder cancer; N0, N1—3-lymph node metastasis; M0, M1—distant metastasis; Q—quartile.

**Table 3 cancers-14-01492-t003:** Univariate and multivariate analysis of overall survival.

Viable	Univariate Analysis			Multivariate Analysis		
	RR	95% CI	*p*-Value	RR	95% CI	*p*-Value
Age (<70 vs. >70)	0.45	0.17–1.2	0.112	-	-	-
Sex (M vs. F)	0.64	0.19–2.13	0.47	-	-	-
Stage (T1 vs. T2–T4)	0.16	0.06–0.4	0.0001	0.36	0.02–8.55	0.53
Grade (low vs. high)	0.17	0.06–0.45	0.0003	0.85	0.14–5.26	0.86
Invasiveness (NMIBC vs. MIBC)	0.13	0.05–0.34	0.00004	0.65	0.06–6.81	0.72
Lymph node metastasis (N0 vs. N1–3)	0.26	0.11–0.63	0.003	1.1	0.21–5.74	0.91
Distant metastasis (M0 vs. M1)	0.17	0.07–0.42	0.0001	0.35	0.08–1.56	0.17
Tumor size (<3 cm vs. >3 cm)	0.30	0.13–0.72	0.007	0.43	0.12–1.57	0.2
Recurrence (Y/N)	0.35	0.05–2.5	0.295	-	-	-
Progression (Y/N)	22	6.08–79.48	0.000002	7.96	1.48–42.5	0.015
CDK9 (low vs. high)	2.7	0.93–7.82	0.06	-	-	-

**Table 4 cancers-14-01492-t004:** Baseline characteristics of TGCA (*n* = 406) cohort.

Clinical Data		*n* (%)
Age (years)		68.1 (range 34–90)
Median follow-up time		1.44 years
Sex	male	299/406 (73.65%)
	female	107/406 (26.35%)
Stage	I	2/406 (0.49%)
	II	129/406 (31.77%)
	III	140/406 (34.48%)
	IV	133/406 (32.76%)
Disease course	Alive	227/406 (55.91%)
	Dead	179/406 (44.09%)

**Table 5 cancers-14-01492-t005:** Univariate and multivariate analysis of overall survival in the TCGA cohort.

Variable	Univariate Analysis			Multivariate Analysis		
	RR	95% CI	*p*-Value	RR	95% CI	*p*-Value
Age (<70 vs. >70)	0.63	0.47–0.85	0.002	0.64	0.48–0.86	0.003
Sex (M vs. F)	1.16	0.83–1.6	0.38	-	-	-
Stage (T1 vs. T2–T4)	0.46	0.32–0.67	0.00004	0.48	0.33–0.69	0.00008
CDK9 (low vs. high)	1.61	1.11–2.33	0.01	1.60	1.1–2.33	0.014

## Data Availability

The data presented in this study are available on request from the corresponding author. The data are not publicly available due to ethical restrictions. Publicly available datasets were analyzed in this study. This data can be found here: https://v21.proteinatlas.org/ENSG00000136807-CDK9/pathology/urothelial+cancer/ (accessed on 10 November 2021) [24].

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
