# Peer review of "The Prognostic Role of CDK9 in Bladder Cancer"

_cancers, 2022, doi:10.3390/cancers14061492_

Round 1

Reviewer 1 Report

authors done extensive improvements

would suggest accepting the paper

Author Response

Thank you for your response. We spell-checked and proofread the manuscript, which was then checked by a native speaker. We adjusted the figures as well.

Reviewer 2 Report

Borowczak et al. submitted an interesting paper for the second time regarding CDK9 expression in urothelial carcinoma. In comparison with the first submission, the paper has improved due to the following changes:

  • Inclusion of quartiles in survival
  • Multivariate analyses
  • Images of CDK9 expression

Prior to acceptance, the figures need additional revision to be suitable for Cancers

  1. I asked for box plots combined with individual values. Right now, the median is plotted as a single dot. I would like to see one dot per patient as well as the boxplot in the same graph. You might need to utilize a different program than Excel.
  2. The quartile data is very important and a nice addition. I suggest you you use thicker lines, these are too thin to be clearly visible
  3. All axis labels and numbers within the figures are too small. Please increase by at least 2pt

Author Response

Thank you for your comments! We considered all of them and revised the manuscript accordingly. 

  1. I asked for box plots combined with individual values. Right now, the median is plotted as a single dot. I would like to see one dot per patient as well as the boxplot in the same graph. You might need to utilize a different program than Excel.
  2. The quartile data is very important and a nice addition. I suggest you you use thicker lines, these are too thin to be clearly visible
  3. All axis labels and numbers within the figures are too small. Please increase by at least 2pt

A1. Previously, we misunderstood the advice. This time figures 2a-c were adjusted - we combined the boxplots with single dots plots and enlarged the markers to keep the charts legibility.

A2. All lines in figures regarding survival analysis has been thickened (0.68 → 1,415 points; figures 3a-b, 4a-b). 

A3. Depending on the figure, the font size has been increased by 2 to 4 points.

In the end, the manuscript was spell-checked and proofread by a native speaker. 

Reviewer 3 Report

The results presented in the manuscript titled 'The prognostic role of CDK9 in bladder cancer' is interesting. It is well-known that over expression of CDK9 results in formation of anti-apoptotic proteins like Mcl-1. It is interesting that over expression of CDK9 helps people to live longer in bladder cancer patients than low expression of CDK9. It would be beneficial to the manuscript if authors could provide plausible explanation for such effects. The results and conclusions were supported by experimental data. 

However, its hard to read the manuscript in track changes mode and most of the time is wording in repeated like '34 were diagnosed with' and some time () put without text. 

Author Response

Dear Reviewer, thank you for your suggestions! We discussed them and revised the manuscript accordingly. 

C1:The results presented in the manuscript titled 'The prognostic role of CDK9 in bladder cancer' is interesting. It is well-known that over expression of CDK9 results in formation of anti-apoptotic proteins like Mcl-1. It is interesting that over expression of CDK9 helps people to live longer in bladder cancer patients than low expression of CDK9. It would be beneficial to the manuscript if authors could provide plausible explanation for such effects. The results and conclusions were supported by experimental data. 

A1:After your review, we searched for a possible explanation of this phenomenon. It seems that a common occurrence in urothelial carcinoma oncogenesis, a loss of chromosome 9, can affect the expression of CDK9. CDK9 gene is located on 9q34 and both deletions of the whole 9q or 9q34 point mutation can impact the functionality of CDK9 and CDK9-mediated proteins. We added an appropriate paragraph in the discussion (part 4.3; “The pathways between CDK9, p53, and other tumor-suppressive proteins(...); page 13 in track-change mode).

C2:However, its hard to read the manuscript in track changes mode and most of the time is wording in repeated like '34 were diagnosed with' and some time () put without text.

A2:  We spell-checked and proofread the manuscript to remove repetitions and unnecessary marks. The manuscript was then checked by a native speaker. Furthermore, the numbers/axis labels of all figures have been enlarged to make them more legible.

Round 2

Reviewer 3 Report

The authors adequately addressed to improve quality of the manuscript. I recommend this manuscript for publication.